# Genome-Wide Identification of *bHLH* Transcription Factor in *Medicago sativa* in Response to Cold Stress

**DOI:** 10.3390/genes13122371

**Published:** 2022-12-15

**Authors:** Guangjun Li, Lei Jin, Song Sheng

**Affiliations:** College of Forest, Central South University of Forestry and Technology, Changsha 410004, China

**Keywords:** *bHLH*, *Medicago sativa* L., cold stress, expression profiling, transcription factor binding sites (TFBS)

## Abstract

Alfalfa represents one of the most important legume forages, and it is also applied as an organic fertilizer to improve soil quality. However, this perennial plant is native to warmer temperate regions, and its valuable cold-acclimation-related regulatory mechanisms are still less known. In higher plants, the *bHLH* transcription factors play pleiotropic regulatory roles in response to abiotic stresses. The recently released whole genome sequencing data of alfalfa allowed us to identify 469 *MsbHLHs* by multi-step homolog search. Herein, we primarily identified 65 *MsbHLH* genes that significantly upregulated under cold stress, and such *bHLHs* were classified into six clades according to their expression patterns. Interestingly, the phylogenetic analysis and conserved motif screening of the cold-induced *MsbHLHs* showed that the expression pattern is relatively varied in each *bHLH* subfamily, this result indicating that the 65 *MsbHLHs* may be involved in a complex cold-responsive regulatory network. Hence, we analyzed the TFBSs at promoter regions that unraveled a relatively conserved TFBS distribution with genes exhibiting similar expression patterns. Eventually, to verify the core components involved in long-term cold acclimation, we examined transcriptome data from a freezing-tolerant species (cv. Zhaodong) in the field and compared the expression of cold-sensitive/tolerant subspecies of alfalfa, giving 11 bHLH as candidates, which could be important for further cold-tolerance enhancement and molecular breeding through genetic engineering in alfalfa.

## 1. Introduction

Low temperature is one of the major abiotic stresses that seriously affects the growth and development of plants, eventually damaging crop production and quality in many plant species [1,2]. Within the last 3 decades, global warming has led to large regional extreme temperature variations. In Europe and Asia, late-spring frosts were estimated to increase to ~35% and ~26%, respectively [3]. “Cold day” occurrences, with temperatures below a certain threshold value, have clearly increased [4]. In this scenario, understanding how plants respond to cold stress is important for further cold-stress-tolerance improvement in crops. Indeed, plants have evolved complex and diverse systems to improve their cold-stress resilience. For harsh cold habitats, signal perception is initiated by a cold-induced transient increase in cytosolic Ca^2+^ concentrations managed by Ca^2+^-permeable channels and detected by numerous receptor proteins that can recognize and interrupt these changes [5,6]. Further, the primary cold-stress signal modifies second messengers or other signal molecules, such as cyclic nucleotides (cAMP and cGMP), amino acids (glutamate and methionine), and various reactive oxygen species, which triggers transcription factors (TFs) to activate the cold-acclimation regulatory networks at the transcriptome level [7,8]. For long-term cold acclimation, plants shift from growth to dormant phases or change morphological characteristics to synchronize with climate rhythms [5,6].

The basic helix-loop-helix (bHLH) transcription factor family has been widely reported in plants, fungi, and animals with pleiotropic regulatory functions [9,10]. In plants, bHLH proteins have been reported to be involved in responses to various abiotic stresses, including drought, cold, salinity, and iron deficiency reviewed by [11,12]. Under low-temperature stress, the AtICE1/AtbHLH116 protein participates with the DREB1/CBF signaling pathway with affect transcription initiation of CBF by promoter region binding; simultaneously, overexpression of AtbHLH116 enhances cold tolerance [13,14]. Similar functions of ICE1/bHLH116 homologs have been observed in other plant species, such as *Dimocarpus longan Lour*, *Pyrus ussuriensisa*, *Brassica campestris*, and *Zoysia japonica* [15,16,17,18]. However, interspecific differences were still obtained, such that PuICE1 requires the interaction with PuHHP1 to further the transcriptional expression elevation of PuDREBα, and even the homolog in rice, OrbHLH001, is independent of the CBF/DREB1 cold-response pathway [16,19]. In addition to ICE1/bHLH116, other bHLH proteins including VaICE2, PtrbHLH, FtbHLH2, MdbHLH3, and Pavb1/18/28/60/61/65/66 are shown to be involved in cold-stress response [20,21,22,23].

Alfalfa (*Medicago Sativa* L.) is one of the most important legume forages, improving soil quality and providing nitrogen as a fertilizer from symbiotic nitrogen fixation. However, significant economic losses result from “late-spring freezing” and other unusual sudden temperature changes that require practical improvement for cold acclimation [24]. To date, a number of TF families have been characterized in alfalfa, such as *bZIP*, *MADS*, *WRKY*, and *DREB* families [25,26,27,28]. However, the *bHLH* family has not been characterized, which is critical for further research on the cold-acclimation mechanism. Thus, we report basic information of *MsbHLH* by homolog search and identified cold-responsive *bHLHs* with cold-stress transcriptome data. Phylogenetic and conserved motif analyses were further processed. In addition, we analyzed the enrichment of TFBSs at the promoter region. Eventually, we examined the transcriptome data of a freezing-tolerant species (cv. Zhaodong) and unraveled 18 certain core *MsbHLH* genes that may be involved in cold/freezing acclimation.

## 2. Materials and Methods

### 2.1. Flow Diagram of Genome-Wide Identification of MsbHLHs in M. sativa

An overview of the study’s workflow is shown in Figure 1. The stepwise identification processes include homolog searching and 2-time RNA-seq analysis (cold response and freezing acclimation). Genetic characterization was performed on 65 cold-responsive *MsbHLHs*. Details are provided below.

### 2.2. Multi-Step Homolog Search of bHLH Genes in Alfalfa

For comprehensive identification and analysis of *bHLH* gene family in alfalfa, sequences of putative *bHLH* genes were obtained from National Center for Biotechnological Information (NCBI, https://www.ncbi.nlm.nih.gov/) GenBank and the Arabidopsis International Resource (TAIR, https://www.arabidopsis.org/, accessed on 1 January 2020) via BLAST search. The draft genome data and annotation information of alfalfa cultivar “XinJiangDaYe” were downloaded from the figshare data repository (https://figshare.com/projects/whole_genome_sequencing_and_assembly_of_Medicago_sativa/66380, accessed on 1 January 2020). *MsbHLH* genes were first searched with TBtools “BLAST GUI Wrapper” [29] to obtain candidate *bHLH* family members. Second, the HMM (Hidden Markov Model) profile PF00010 (bHLH domain) was retrieved from the Pfam database (https://pfam.sanger.ac.uk/, accessed on 1 January 2020) to identify the putative *MsbHLH* genes from previous BLAST filtered candidates. Basic parameters including molecular weight (MW) and isoelectric point (*p*I) were predicted using the ProtParam tool (https://web.expasy.org/protparam/, accessed on 1 January 2020) [30].

### 2.3. Identification of Cold-Responsive MsbHLHs with RNA-Seq Data

The raw RNA sequencing data were obtained from NCBI Sequencing Read Archive (SRA, https://www.ncbi.nlm.nih.gov/sra, accessed on 1 January 2020) through the SRA toolkits “prefetch” (version 2.8.0). Project “SRP144299” was downloaded for cold-stress analysis [31], “SRP060503” was used for cold-acclimation study [32], and “SRP055547” was used for cross-validation of the cold-acclimation-related result [33]. Raw data (raw reads) in “fastq” format were first qualified with FastQC program [34] for Q20, Q30, GC-content, and sequence duplication level; the data were then processed in Hisat2 version 2.2.1 [35] for read alignment to the alfalfa genome “XinJiangDaYe” [29]. The reads were subjected to fragments per kilobase of transcript per million fragments mapped (FPKM) conversion to obtain the expression value of genes and transcripts. In-house R scripts were used to analyze gene expression and generate heatmaps. Heatmap was generated using ggplot2, reshape2, gplots, and dplyr packages in R version 4.1.2 [36]. Source code in R and RNA-seq data were provided in Appendix A.

### 2.4. Phylogenetic, Gene Structure, and MEME Conserved Motif Analysis

With the aim to analyze the sequence features of cold-responsive *bHLH* genes in alfalfa, 170 bHLH proteins, including 105 from A. thaliana and 65 pre-selected MsbHLHs from *M. sativa*, were loaded on MEGAX version 10.2.6 (https://www.megasoftware.net, accessed on 22 March 2016) using default “ClustalW” parameters for alignment. The phylogenetic tree was then constructed with default “Neighbor-joining” algorithm parameters. Phylogenetic trees were then visualized by iTol (https://itol.embl.de/, accessed on 1 October 2020) [37]. Conserved motifs of MsbHLHs proteins were predicted using the MEME suite (version 5.4.1, https://meme-suite.org/meme/index.html, accessed on 25 August 2021) [38] with default parameters besides the “motifs should find”, which was set to 10. The coding sequences (CDS) and the structure of all genes were graphically displayed with TBtools function “Gene Structure View”. Gene structure info from “GFF” file containing the predicted gene structure annotation was obtained from “figshare.com, accessed on 17 January 2012” described above.

### 2.5. Putative TFBSs Analysis in the Promoter Regions of MsbHLH Genes

Identification of transcription factor binding sites (TFBS) is important for understanding the function of TFs and their interactions with TF-TFBS pairs. TFBSs were analyzed to determine the transcription levels of their downstream genes. A total of 2 kb of upstream sequence from starting codon was analyzed for each *bHLH* gene by PlantPAN 3.0 webserver “multiple promoter analysis” tool (http://plantpan.itps.ncku.edu.tw/promoter_multiple.php, accessed on 17 January 2012) [39]. Source code in R and TFBS data from PlantPAN3.0 were provided in Appendix A.

## 3. Results

### 3.1. Genome-Wide Identification of MsbHLH Genes in M. sativa

To identify the *MsbHLH* genes in alfalfa, a local BLAST program was performed through TBtools with the cultivar “XinJiangDaYe” genome sequence [31], and a homolog search with 105 putative *Arabidopsis bHLH* genes together with conserved domain HMM screening using PF00010 (bHLH domain) identified a total of 469 bHLH candidates. These MsbHLH genes range from 92 to 1396 amino acids (average/median 333/309 aa). The predicted molecular weights (Mw) of the MsbHLH proteins ranged from 10.35 kDa to 159.5 kDa, and the isoelectric points (*p*I) ranged from 4.44 to 11.31 (Appendix A).

### 3.2. Identification of 65 Cold-Stress-Induced MsbHLHs with Transcriptomic Data

The *bHLH* family is one of the most extensively researched superfamilies that participates in pleiotropic regulatory roles in response to abiotic stresses, such as drought, cold, heat, and salinity stresses, and the size of this family is the second largest in plants [reviewed by 11–13]. Thus, with the aim to identify the cold-responsive components from a total of 465 *MsbHLHs*, a publicly available RNA-seq dataset obtained from NCBI related to time-series (0, 2, 6, 12, and 24 h) cold-stress treatment (4 °C) of alfalfa variety Zhongmu No.1 was analyzed [31]. The expression level of these genes was presented in a heatmap (Figure 2). With the threshold of “FPKM > 0.1”, “log_2_ fold change > 1”, and “*p*-value < 0.05” to filter the cold-induced *MsbHLH* genes, 65 out of 469 *MsbHLHs* were identified and clustered into six clades (Figure 2), which consist of 12 (Clade 1), 18 (Clade 2), 11 (Clade 3), 9 (Clade 4), 12 (Clade 5), and 3 (Clade 6) genes, respectively. In general, genes in Clades 1, 2, 5, and 6 were induced by short-term (less than 24 h) cold stress while genes in Clades 3 and 4 were induced by a relatively longer (more than 1 day) cold stress (Figure 2).

### 3.3. Phylogenetic, Gene Structure, MEME Motifs of 65 Cold Responsive MsbHLHs

To explore the distribution and structural diversification of conserved motifs of MsbHLH proteins, we analyzed the conserved domains of MsbHLHs by the MEME (Multiple Em for Motif Elicitation) online tool (Figure 3). With only modifying the default setting of “motif numbers” from 3 to 10, we observed that none of the 65 MsbHLHs contain all 10 motifs while sharing the conserved bHLH domain (motif 2, yellow). The motif diversity was relatively conserved in each subfamily (Figure 3A). In addition to motif 2, motifs 4, 6, 8, 9, and 10 were less observed while motif 1 was almost characterized in all genes except “Ms.gene055887.t1” and “Ms.gene63249.t1”. Gene structure analysis showed that the number of exons in *MsbHLH* genes varied from 1 to 18 (Figure 3B). “Ms.gene035172.t1” contains the relatively highest number of 18 exons and has the largest intron regions. Furthermore, we integrated the phylogenetic and transcriptomic results to facilitate insight into the potential functional diversity of *MsbHLH* genes (Figure 4). In the current study, 65 *MsbHLH* genes were previously clustered into six clades according to their cold response gene expression level (Figure 2). However, clustered genes were relatively grouped separately with phylogenetic analysis according to their protein sequences; for example, genes in Clade 1 were mostly grouped at the lower part while those in Clade 2 were grouped at the upper part in this circle phylogenetic tree (Figure 4).

### 3.4. TFBS Analysis of Cold-Responsive MsbHLH Genes

With the aim to further unravel the potential regulatory mechanism of *MsbHLH* genes in the complex cold-related networks, the promoter region (−1 to −2000) of each *MsbHLH* was uploaded to the PlantPAN webserver [38]. In total, more than 600 TFBSs were characterized at the promoter region. To statistically investigate whether there were differences in TFBS composition, we initially established a baseline of TFBS, which is dissimilar to the TFBS studies in other studies that only calculate differentially enriched TFBS above a certain number [40,41,42]. In the current study, the TFBS composition of each gene was divided by baseline data and then followed with log_2_-fold change calculation, and the final statistical results were clustered by “heatmap.2” from “gplot” in R (Figure 5). Obviously, genes showing similar expression patterns were clustered together, such as Clades 2 and 5, Clades 1 and 6, and Clades 3 and 4 (Figure 5), respectively. This result indicates a relatively highly conserved TFBS that regulates gene expression in response to cold stress. To more intuitively explore the promoter regions, we analyzed the top significantly differentially enriched TFBS of each clade, which gives a total of 59 TFBSs belonging to 15 families (Appendix A), including TFBS binding to the TF family of AP2, TCP, bZIP, MADS box, MYB.

### 3.5. Transcriptome Analysis of 65 Cold-Responsive MsbHLHs Identified 18 Genes Related to Overwinter Cold Acclimation in M. sativa cv. Zhaodong

*M. sativa* cv. Zhaodong is a major forage legume cultivar that can survive in northern China, where temperatures may reach around −30 °C in winter. Song et al. (2016) provided a systemic analysis of adaptive mechanisms for frost tolerance using RNA-seq [32]. *M. sativa* cv. Zhaodong was cultivated in September and harvested in mid-winter after freezing adaptation during a two-month temperature decrease. In this adaptation process, the antioxidant defense system was believed to confer freezing tolerance in a rapid way, and numerous potential freezing-sensing and signal transduction components were revealed. Based on the transcriptome data, we identified 18 *MsbHLHs* that contribute to long-term (much longer than 1-day cold stress) cold adaptation and alfalfa survival throughout the winter (Figure 6). Interestingly, these upregulated genes are not mainly distributing in specific clades while almost grouped separately. This finding suggests that a complex regulatory network is consistent with the development of cold tolerance.

### 3.6. Expression Difference between M. sativa Subspecies during Plant Development

To cross−validate the previously identified cold-responsive *MsbHLH* genes, a de novo transcriptome data assembly from six different tissues of two alfalfa subspecies (*M. sativa* ssp. *sativa* from the Middle East and *M. sativa* ssp. *falcata* from colder Central Asia) was examined. In this study, we comprehensively examined the expression level of 18 *MsbHLHs* in specific tissues including root, leaf, flower, elongation stem internode, post-elongation stem internode, and nitrogen-fixing nodule (Figure 7). These results clearly indicate a distinct and ecosystem-dependent expression pattern of 11 *MsbHLHs* between *falcata* and *sativa*. Genes 96814, 74305, 69938, 54951, 31079, 09877, and 006766 have significantly lower expression levels in cold-tolerant *falcata* across all six tissues. Genes 60418, 46828, 49862, and 035172 have higher expression levels in more than half of tissues (Figure 7). Thus, we would like to suggest that the 11 *MsbHLHs* would be interesting molecular targets for further cold-stress improvement in alfalfa.

## 4. Discussion

The *bHLH* family, which has emerged as the second-largest (less than MYB family) TF family in plants, is essential for regulating plant growth in response to environmental stresses [43]. Although the *bHLH* family has been studied in many plants, relatively little information is available regarding alfalfa *bHLH* genes. In the current study, we identified 469 genes encoding *bHLH* transcription factors that had analogous structural characteristics to those in *Arabidopsis* and contain the bHLH domain in the recently released alfalfa genome [29]. Freezing temperatures are a major factor affecting alfalfa development and reducing production and survival, particularly in China’s high-altitude and northern locations [32]. Meanwhile, *bHLH* transcription factor plays varied significant regulatory role for plant growth and development as well as tolerance to abiotic stresses including drought, salt, and low temperature [44]. To focus the cold-responsive components, we primarily analyzed *MsbHLH* genes with cold stress RNA-seq data that give 65 cold-induced *MsbHLHs* (log_2_-fold change > 1, *p*-value < 0.05, and FPKM > 0.1) (Figure 2). In general, variation in gene family members’ expression profiles could be a reflection of the inherent variety of functions [45]. In addition, we found that 65 *MsbHLHs* were clustered into six clades based to their expression patterns (Figure 2), indicating that distinct clades may engage in the cold-stress response at different stages. Indeed, cold response includes perception, signal transduction, and the regulation of gene expression. Short-term cold stress resulted in rapid and transient increases in PLD/PA, proline, and reaction oxygen species (ROS) in leaves, but these small signal molecules were reduced after long-term cold stress. Interestingly, the opposite result of proline synthesis and ROS signals was observed in the roots [46]. Similar results have been observed in 47 cold-responsive *OfbHLHs* (osmanthus fragrans) that show various expression patterns with a range of 0.5 (short) to 120 (long) h cold stress [47]. To date, the mechanism of how these small molecules interact and/or coordinate to *bHLH*-related signal transduction pathways is little known [48]. Thus, more specific and in-depth experiments with the *bHLH* family still need to proceed.

Analysis of gene structures provides important information on phylogenetic relationships. In Figure 3, we observed that the number of exons and conserved motif diversification shows less variation in *bHLH* subfamilies while the number of exons in *MsbHLHs* ranged from 1 to 18, and similar results have also been obtained in conservative motif arrangements in which gene motifs are consistent within subfamilies but varied ranging from 1 to 7. This indicates loss or insertion during the evolution of *M. sativa*. In general, genes with the same functions were clustered into the same subfamily. However, the phylogenetic tree and RNA-seq data integration show that there are some relatively diverse expression differences of *MsbHLH* within subfamilies (Figure 4). This is similar to *OfbHLHs*, which have genes with high sequence similarity showing similar expression patterns in response to short/long terms of cold stress but contain several outliers, such as *OfbHLH117*/*120* in subfamily IIIB, and only *OfbHLH194* was downregulated at 0.5 h in this family [46]. We made the assumption that genes in each subfamily generally perform comparable biological functions while being activated or controlled by various upstream transcription factors, which causes the genes to engage in various stages of cold response. In this scenario, we further analyzed the transcription factor binding sites (TFBS) of the promoter region (−1 to −2000 bp) of 65 *MsbHLH* genes.

The TFBSs in the promoter regions regulate gene expression. In this study, we generated the TFBS frequency matrix based on the number of predicted TFBSs in the promoter region of 65 *MsbHLH* genes. In general, genes showing similar expression patterns were grouped, such as Clades 2 and 5, Clades 1 and 6, and Clades 3 and 4 (Figure 5). However, these TFBSs are differentially enriched between different groups (Appendix A). For instance, the AP2/ERF family was mostly enriched in Clade 1, 2, 5, and 6, in which genes were induced by short-term cold stress that may lead to a rapid response, and DREB genes belonging to the AP2/ERF family are importantly involved in short-term cold-stress response and further cold acclimation [reviewed by 8]. In addition, TFBSs belonging to other families, such as TCP, MYB, HSF, play an important role during plant growth and are involved in stress response. The composition differences among 65 *MsbHLHs* indicate that the regulation of genes that caused the following stepwise gene regulation is well-established. However, the underpinning mechanisms of how the cold-stress signal was transduced from primary ROS Ca^2+^ sensors to *bHLH* and even to downstream genes are still little known.

Over the past two decades, numerous proteins in the *bHLH* family have been identified as being intricate in the tolerance to low-temperature stress [reviewed by 11–13]; for example, *ICE1*, a typical MYC-type *bHLH* transcription factor, can regulate cold-responsive signal transduction [49,50]. Simultaneously, ICE1 positively regulates *CBFs* expression and freezing tolerance by directly binding to the *CBF3* promoter region [14]. We previously characterized 65 *MsbHLHs* that may be involved in short-term cold stress sensing or signal transduction. However, as an important perennial forage, overwinter cold-tolerance formation is essential for alfalfa to avoid or decrease devastating freezing damage [51]. In northern China, *M. sativa* cv. Zhaodong is a native cold-tolerant species that can survive temperatures as low as −30 °C, and the overwinter rate is more than 90% [51]. Thus, a long-term transcriptome analysis of adaptive mechanisms for frost tolerance of cv. Zhaodong was investigated [32]. Here, we focused on the expression pattern of 65 previously identified cold-tolerance-related *MsbHLHs* (Figure 6). The result indicates that 18 *MsbHLHs* were induced in winter and these genes may participate both in short- (1 h to 48 h, 4 °C) and long-term (3 months, maximum to −30 °C freezing) low-temperature response. Moreover, the cross-validated RNA-seq result of cold-sensitive/tolerant *M. sativa* subspecies unraveled 11 out of 18 MsbHLHs, grouped with seven showing lower expression level in *falcata* (cold tolerance) and four in *sativa* (cold sensitive) [33]. In addition, these genes may provide promising molecular targets for the improvement of cold tolerance in alfalfa (Figure 6 and Figure 7).

## 5. Conclusions

In this study, we have genome-widely identified 469 putative *MsbHLH* genes using recently released whole-genome sequence datasets, characterized in detail by their genetic lineage and gene structures as well as coding proteins that consist of a conserved bHLH domain. In particular, 65 out of 469 differentially expressed *MsbHLHs* were characterized as potential cold-stress-responsive components, which may serve as crucial trans-activators participating in or controlling a signal transduction network(s) regulating growth adaption to cold stress. Moreover, 11 *MsbHLHs* were cross-validated as genes involved in long-term tolerance to freezing environments, and these *MsbHLH* were differentially expressed in cold-sensitive/tolerant *M. sativa* subspecies and also significantly accumulated after long-term freezing treatment. In future studies, it may be interesting and necessary to functionally appreciate such *MsbHLH* genes at molecular and physiological levels. Thus, we believe that our findings with adding many new members to the *bHLH* family in alfalfa should provide valuable information for further profoundly understanding *bHLH*s’ action in plant cold-stress behavior, assisting us in taking a proper approach toward aiming at improving crop cold tolerance.

## Figures and Tables

**Figure 1 genes-13-02371-f001:**
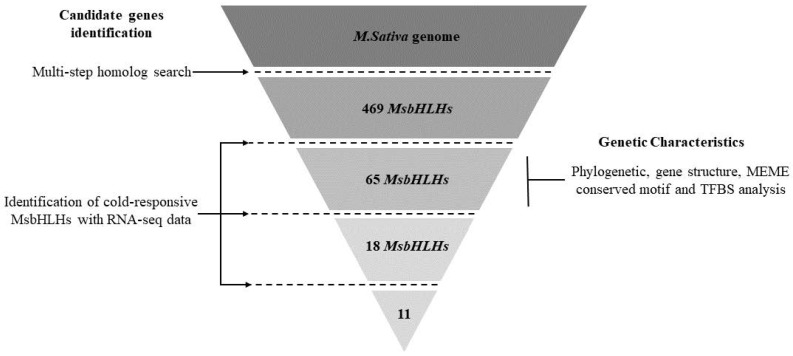
Flow diagram of genome-wide identification of bHLHs in *M. sativa*. The dashed lines represent the identifying procedures that incrementally eliminate the potential genes involved in both cold acclimatization and cold response.

**Figure 2 genes-13-02371-f002:**
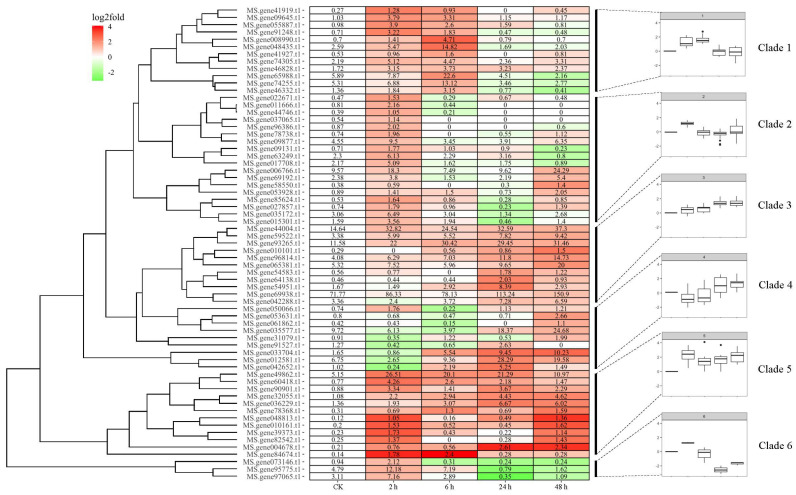
Clustering−approach−based heatmap of *MsbHLH* genes. The heatmap represents the expression pattern of genes in response to cold stress. Relative expression levels were calculated as a Log_2_-fold change against CK (see Section Materials and Methods). The red color shows an upregulation of a given gene, and the green indicates a downregulation. Labeled number in each tile is the expression level “FPKM”.

**Figure 3 genes-13-02371-f003:**
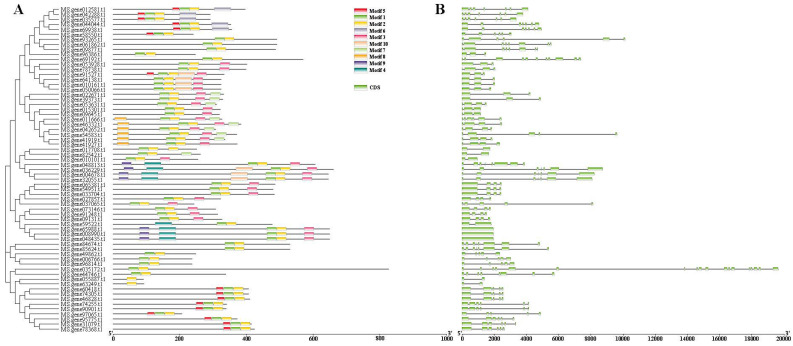
Analysis of the conserved motifs (**A**) and gene structure (**B**) of 65 cold-stress-responsive *MsbHLH*s.

**Figure 4 genes-13-02371-f004:**
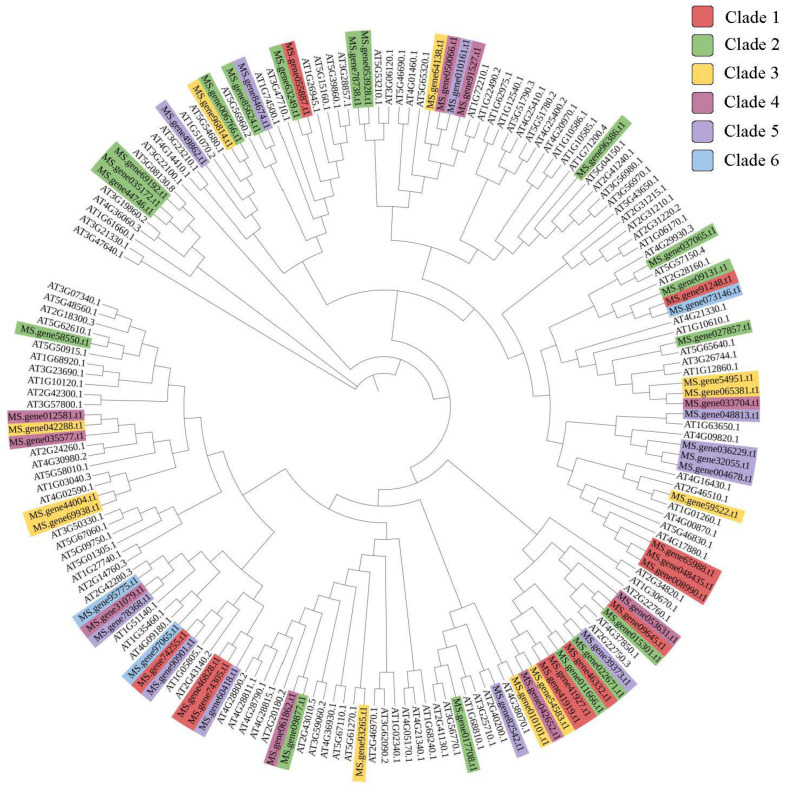
Phylogenetic tree of bHLH proteins from Arabidopsis thaliana and Medicago sativa L. Evolutionary relationships were constructed using the neighbor-joining (NJ) method (see Section Materials and Methods). The shaded color indicates the genes from different clades that are grouped by gene expression pattern under cold stress.

**Figure 5 genes-13-02371-f005:**
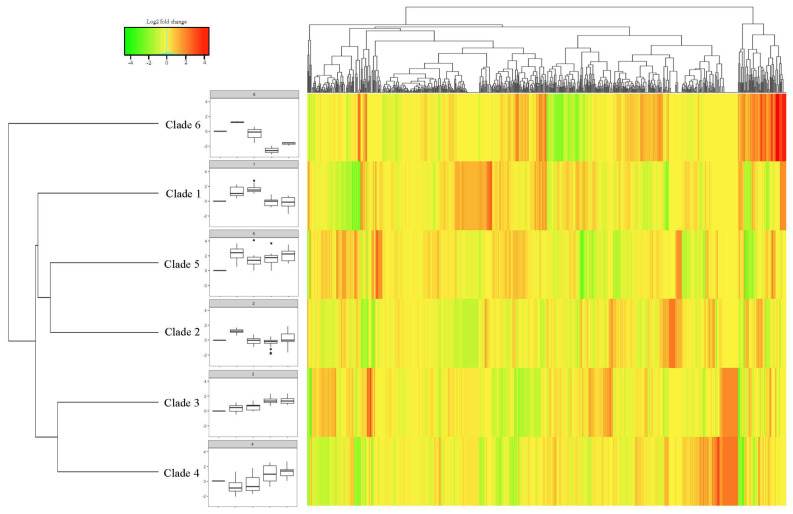
Promoter analysis of 65 cold-stress-responsive *MsbHLH* genes by PlantPAN. TFBSs of 65 cold-stress-responsive *MsbHLHs* of the −2000 bp 5′ upstream regions are statistically analyzed. The ratio of each TFBS was divided with the baseline ratio, and then the heatmap was generated by heatmap.2 from Gplots in R. Clades 1–6 are related to the gene expression patterns of *MsbHLHs* shown in Figure 2.

**Figure 6 genes-13-02371-f006:**
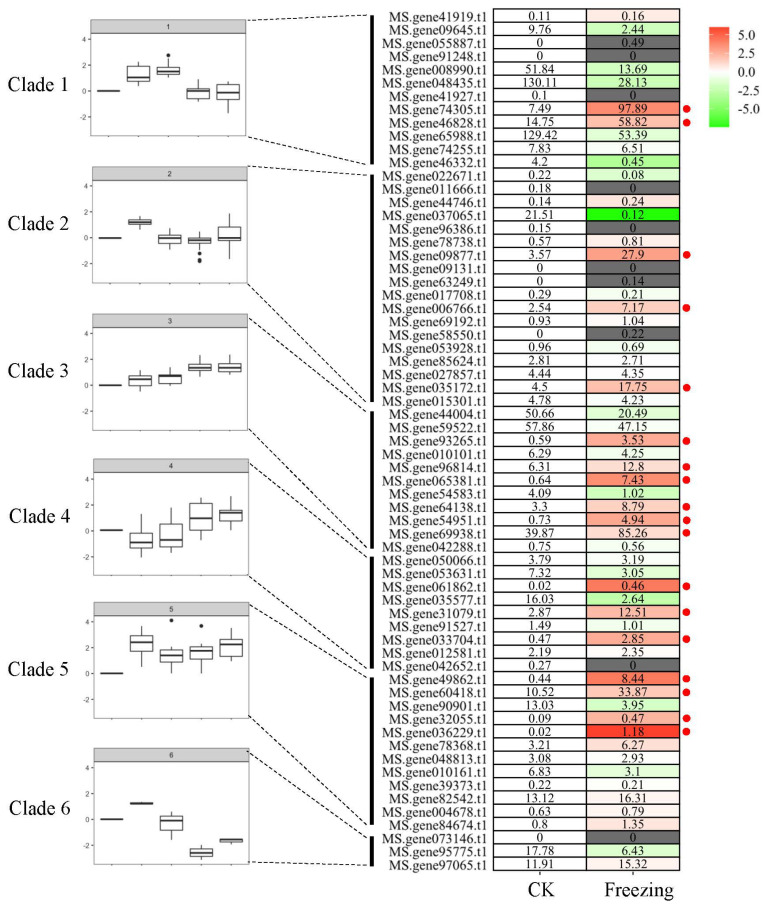
Clustering-approach-based heatmap of *MsbHLH* genes in freezing acclimation alfalfa. Red point represents the significantly upregulated *MsbHLH* genes. The red color shows an upregulation of a given gene, and the green indicates a downregulation. Labeled number in each tile is the expression level “FPKM”.

**Figure 7 genes-13-02371-f007:**
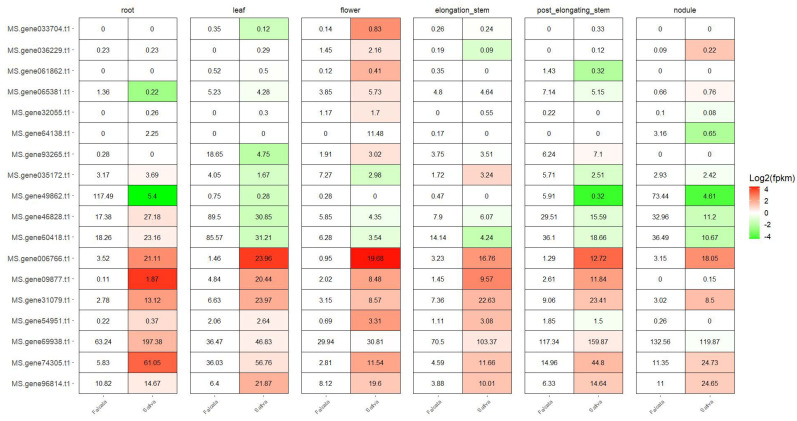
The expression level of MsbHLH genes in different tissues of sensitive (*M. sativa* spp. *falcata*) and tolerant (*M. sativa* spp. *Sativa*) cultivars. Relative expression levels were calculated as a Log_2_−fold change against CK (see Section Materials and Methods). The red color shows an upregulation of a given gene, and the green indicates a downregulation. Labeled number in each tile is the expression level “FPKM”.

## Data Availability

RNA-seq data obtained in this study are available in NCBI, and others are presented in this study and its Appendix A. The source code is available from the corresponding author on reasonable request.

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
