# Peer review of "Genome-Wide Identification of bHLH Transcription Factor in Medicago sativa in Response to Cold Stress"

_genes, 2022, doi:10.3390/genes13122371_

Round 1

Reviewer 1 Report

enclosed

Author Response

Dear Reviewer,

Thank you very much for your suggestions. The goal of this study is to gather information about bHLH in alfalfa and identify cold-tolerance regulatory genes. As a result, we followed the majority of your suggestions but did not go into great detail about the interaction of bHLH with other TFs. The following are our corrections:

Abstract

Comment 1: The comparison with freeze tolerant species is what you mention, but have no information neither about the post-transcription regulators, nor do u look up for the proteome data for validation of hypothesis where you are giving the bHLH transcriptomes as key. Either you could provide an insight with documentary evidence in supplementary evidences and the data provided could provide the insight into the exciting finding. In conclusions, please summarize coherently the output of last couple of lines of abstract. Needless to say in discussion if you could highlight to emphasize clearly the other transcription factors associated with cold stress and superiority of the bHLH transcription factor it would be useful.

Dear editor and reviewers, validating the result with proteomics data is an excellent suggestion; however, only one paper analyzing the proteomic of alfalfa under cold stress was found, and it was published in 2015 without the related omics data released. In this case, we attempted to use another dimension of transcriptome data involving 6 tissues of cold sensitive/tolerance cultivars, which may partially compensate for the lack of proteomics (see Fig. 6). With this additional validation, we identified 11 MsbHLHs that were differentially expressed between almost all tissues of two cultivars, indicating a potential regulatory mechanism that causes cold-tolerance in cold-sensitive cultivars. The lack of proteomics data is an unavoidable reason, and this is where we intend to go in the future.

Introduction

Comment 2: The opening statements in the research article make it rather vague, ideally the second paragraph and an insight into the known information about the various transcription factors associated with cold stress and other post transcription regulators, the proteomic information and also the insight into the morphological detailing would generate interest of the reader.

Comment 3: Please delete 28-38.

Comment 4: 47-58 are incoherent please rewrite and please dress the key points with suitable references with a line of your motivation for selection of problem reported herein.

For comment 3-5, we deleted lines 28-38 and made some changes based on the suggestions; for more information, please see the corrected version.

Materials and Methods

Comment 5: Since the strategy used in materials and methods is a whole lot without justification so it appears incoherent and does not interest the reader, suitable flow diagram addition of the work would make it read very well.

Thanks for the suggestion that is useful, and now we added Fig. 1 to explain the workflow of this study. And even we can provide the source code with R for people who want to reproduce the results.

Results and discussion

Comment 6: In this section the author needs to pause to explain suitably the data with suitable explanation. It is confusing and need to be spruced up.

Following your suggestion, with the addition of MsbHLH cross-validation and the workflow figure, we believe it is now much easier for the reader to understand.

Conclusions

Comment 7: This section needs total recrafting making delivery precise and accurate.

Please see the updated version, we deleted some sentences and try to just make it easy to understand.

Sincerely,

Song Sheng

Reviewer 2 Report

The topic entitled "Genome-wide identification of bHLH transcription factor in Medicago sativa in response to cold stress. Alfalfa (Medicago sativa L.) is a perennial legume widely cultivated to provide high-quality forage in the form of hay, silage and to a lesser extent as a grazing crop, as well as for improving soil fertility.

The availability of alfalfa lines that can be easily transformed with Agrobacterium and regenerated in tissue culture, makes this plant attractive for genetic engineering and provides the opportunity to develop new uses for alfalfa with value-added products of commercial interest.

Alfalfa is the most important forage crop.

English language need minor revision throughout the text of the paper.

Problem, need and scope of the paper is not much clear.

Figures (1,2,3 & 4) are not mentioned in the test.

Discussion part should be more improved with latest references. Some references are incomplete.

Author Response

Dear Reviewer,

Following your suggestions, in this newly submitted version, we added the workflow figure and a cross-validation section. We believe it is now much easier to understand for the reader, and the end result is more solid. Please see the attached Word document for more information.

Sincerely,

Song Sheng
